# Molecular Identification of African *Nymphaea* Species (Water Lily) Based on ITS, *trnT-trnF* and *rpl16*

**DOI:** 10.3390/plants11182431

**Published:** 2022-09-18

**Authors:** Zhi-Hao Qian, Justus Mulinge Munywoki, Qing-Feng Wang, Itambo Malombe, Zhi-Zhong Li, Jin-Ming Chen

**Affiliations:** 1Aquatic Plant Research Center, Wuhan Botanical Garden, Chinese Academy of Sciences, Wuhan 430074, China; 2University of Chinese Academy of Sciences, Beijing 100049, China; 3Sino-Africa Joint Research Center, Chinese Academy of Sciences, Wuhan 430074, China; 4Plant Diversity Research Center, Wuhan Botanical Garden, Chinese Academy of Sciences, Wuhan 430074, China; 5East African Herbarium, National Museums of Kenya, Nairobi 451660-0100, Kenya

**Keywords:** *Nymphaea*, DNA barcoding, ITS, *trnT-trnF*, *rpl16*, species identification

## Abstract

The genus *Nymphaea* L. (water lily) is the most diverse genus in the family Nymphaeaceae, with more than 50 species worldwide, including 11 species distributed in Africa. The complex and variable morphology of *Nymphaea* makes it extremely difficult to accurately identify species based on morphological characteristics alone. DNA barcoding has the potential to identify species accurately. In this study, 158 *Nymphaea* populations from seven African countries were collected for species identification by ITS, *trnT-trnF* and *rpl16*. Additionally, the three candidate DNA barcodes were evaluated for genetic distance and barcoding gap. Based on the comprehensive analysis of sequence similarity, genetic distance method and phylogenetic tree, a total of 137 populations of seven *Nymphaea* species from African were well-identified, including *N. lotus*, *N. petersiana*, *N. zenkeri*, *N. nouchali* var. *caerulea*, *N. micrantha* and *N. guineensis*. ITS has more obvious advantages over *trnT-trnF*, *rpl16* and *trnT-trnF*+*rpl16* in the intraspecific and interspecific variation differences and barcoding gap and can identify most species. *trnT-trnF* and *rpl16* can identify some species that cannot be identified by ITS. The results showed that it is more appropriate to apply the combination of ITS and *trnT-trnF* (or *rpl16*) as the DNA barcoding of *Nymphaea*. Additionally, this study further enriches the DNA barcoding database of *Nymphaea* and provides a reference basis for studying taxonomy, phylogenetics and evolutionary origin of *Nymphaea*.

## 1. Introduction

The water lily (*Nymphaea* L.) is the most species-rich, phenotypically diverse and geographically widespread genus in the order Nymphaeaceae, with about 50 species worldwide [1,2]. The water lily is an aquatic flower with high ornamental value and is deeply loved by people for its colorful flowers, sacred and sublime symbolism and water purification capacity. Additionally, the water lily and its relatives, such as the basal lineage of angiosperms, are important for understanding the early evolutionary patterns of angiosperms [3,4]. Conard [5] divided the genus *Nymphaea* into five subgenera in his monograph, including subg. *Anecphya*, *Brachyceras*, *Hydrocallis*, *Lotos* and *Nymphaea*. These five subgenera were also supported by molecular evidence and are now generally accepted [2]. Although the major genealogies within the genus *Nymphaea* were well-resolved, interspecific relationships within these taxa remain uncertain [6]. Though more than a century since the first monograph of the genus *Nymphaea* was published by Conard [5], discoveries and circumscription of new species of genus *Nymphaea* are still ongoing. A quarter of the currently accepted *Nymphaea* species were completed more than 40 years ago, and the status of many species is unknown, especially subg. *Brachyceras* and *Lotos* [7]. However, the complex and variable morphology of *Nymphaea* species makes their correct identification problematic, and even the morphological characteristics of the same species are diverse [8]. Moreover, the presence of natural hybrid groups in the genus *Nymphaea* increased the difficulty of classical classification [9]. The existing description of the morphology is for the mature plant, while that of the same plant at its youthful stage is unknown. In this case, one is likely to misidentify the species based on its morphology in the early stages of development. Therefore, the accurate identification of genus *Nymphaea* species using morphology alone is difficult, and additional evidence is needed to address the taxonomic complexity of the genus.

Molecular techniques have become increasingly popular in taxonomic studies in the last decades [10,11,12]. DNA barcoding is one of the molecular methods for species identification using one or several standardized DNA sequences [13]. Currently, many DNA barcodes have been tested for identification in plants [14,15,16]. Although a number of DNA markers were assessed as candidate DNA barcodes for plants, no barcode alone performed as well as COI in animals [17]. Each plant DNA barcode has its own strengths and weaknesses, resulting in different success rates for identification among different taxa. Several different combinations of candidate DNA regions were proposed for barcoding plants [18,19,20]. Moreover, the main problems faced in selecting plant candidate DNA barcodes are a slow evolutionary rate, hybridization, gene infiltration or incomplete lineage sorting, which ultimately hinder species identification [17]. Thus, screening for suitable DNA barcodes for different taxa is essential to address taxonomic complexity.

Two subgenera, *Brachyceras* and *Lotos*, are mainly distributed in Africa. According to the Global Biodiversity Information Facility (GBIF), a total of 11 *Nymphaea* species are distributed in Africa. However, this is only based on morphological records and lacks the support of molecular data. Therefore, it is necessary to carry out the DNA barcoding study to help the taxonomic identification of African *Nymphaea* species. Currently, studies have applied the ribosomal DNA (nrDNA) ITS and the chloroplast DNA (cpDNA) *trnT-trnF* to the phylogenetic analysis of the genus *Nymphaea* and suggested that ITS and *trnT-trnF* could be used as potential barcodes for the identification of *Nymphaea* species [7,21]. In this study, we used ITS and two cpDNA regions (*trnT-trnF* and *rpl16*) for a DNA barcoding analysis of African *Nymphaea* species. The present study aimed to identify African *Nymphaea* species using the three DNA barcodes and assess the ability of the three candidate DNA barcodes to identify African *Nymphaea* species.

## 2. Results

### 2.1. DNA Barcodes Universality and Sequence Characteristics

The PCR amplification and sequencing success rates of the three candidate barcodes were 98.1–100% and 96.2–100%, respectively, indicating that these primers had good versatility in *Nymphaea*. The sequences obtained were submitted to the GenBank database, and the accession numbers are shown in Appendix A. The lengths of ITS, *trnT-trnF* and *rpl16* after sequence alignment were 731 bp, 1507 bp and 766 bp, respectively, and the number of parsimony information sites were 196, 70 and 15, respectively. The GC content of the ITS region was 51.8%, which was much higher than that of the two cpDNA sequences. Moreover, the number of variation sites (212) and variation rate (29.73%) of ITS was the highest, followed by *trnT-trnF* (126 and 8.36%, respectively). Therefore, ITS sequences were considered ideal DNA barcoding for *Nymphaea* plants. The sequence characteristics of three DNA barcodes of the above *Nymphaea* are shown in Table 1.

### 2.2. Species Identification of African Nymphaea Species Based on DNA Barcoding Analysis

Species identification in this study employed a similarity-based approach using BLAST. The results showed a high similarity of ITS and *trnT-trnF* with the sequences of the GenBank database by BLAST. The similarity of ITS was above 98.7%, except for the similarity (97.7%) of No. 733, 743 and 761, and the species identified by ITS and *trnT-trnF* belonged to the subg. *Brachyceras* and *Lotos* (Appendix A). Some samples were identified as *N. caerulea* and *N. capensis* by ITS and *trnT-trnF*, respectively. According to the latest GBIF database, *N. caerulea* and *N. capensis* were classified as one species named *N. nouchali* var. *caerulea*. 

We found that some species may have morphological identification errors through homology matching and analysis. For example, No. KISW, MSLK and MTR were identified as subg. *Brachyceras* at the time of field collection, and the results of BLAST indicated that the three samples belonged to subg. *Lotos*. The No. MGB and MWB were identified as *N. heudelotii* in the field, and the results of BLAST showed the two samples were identified as *N. nouchali* var. *caerulea* with a high match (>99%). No. 714, DID, HOH and KES were inconsistent in the subgenera identified by ITS and *trnT-trnF*, and it is possible that the samples were contaminated. Based on a BLAST search, 64 samples identified by ITS, and 50 samples identified by *trnT-trnF* were consistent with morphological classification. Additionally, 109 samples were identified based on ITS and *trnT-trnF* together, of which 63 unknown species were identified.

ASAP is a new method to build species partitions from single locus sequence alignments. When sequences that were in different groups are clustered together, the asap-score is the smaller, the better [22]. The *Nymphaea* species were divided into two groups (asap-score = 2.0) by the ASAP analysis of ITS, corresponding to *Brachyceras* and *Lotos*, and the species could not be successfully identified (Appendix A). For cpDNA sequences, the ASAP analysis of *trnT-trnF*, *rpl16* and *trnT-trnF*+*rpl16* suggested divided into four, two (or three) and three groups, respectively, which also could not distinguish species well (Appendix A). However, all three cpDNA sequences can identify *N. petersiana* based on this method. 

The NJ and BI analyses of the four DNA regions showed similar topologies. The phylogenetic tree constructed based on ITS displayed a good topology structure with reliable monophyletic evolutionary branches (Figure 1). *N. petersiana* and *N. zenkeri* were monophyletic in subg. *Lotos*. While, in subg. *Brachyceras*, some individuals of *N. guineensis*, *N. micrantha* and *N. nouchali* var. *caerulea* were clustered together, and the phylogenetic results could not identify the other species. In addition, the phylogenetic trees of *trnT-trnF*, *rpl16* and *trnT-trnF*+*rpl16* generated similar topologies (Appendix A). The phylogenetic trees using three cpDNA barcodes showed all samples of the subg. *Lotos* were clustered into two major branches, and only all individuals of *N. nouchali* were monophyletic in subg. *Lotos*. Other individuals clustered together exhibited comb structures. The No. 3405 was identified as *N. lotus* and *N. petersiana* based on ITS and cpDNA, respectively. According to Borsch [2], *N. petersiana* had “AGAA”-SSR in the *trnL-trnF* spacer region. Among all the samples obtained in this study, only 3405, KGS, TGN and GMN had “AGAA”-SSRs in the *trnL-trnF* spacer region (Figure 2). Therefore, the clustering results of the three cpDNA sequences for *N. petersiana* were reliable, which is consistent with the identification results of ASAP. Based on the phylogenetic analysis, ITS and three cpDNA sequences, 51 samples were jointly identified, of which 33 unknown samples were identified (Appendix A).

For the combined identification results based on homology searches, the genetic distance method and phylogenetic analysis, 137 *Nymphaea* species were identified in this study, and 7 samples were corrected based on morphological misidentification. Combined with morphological identification, 144 samples of 7 species were finally identified, and 14 samples were identified as failed (Appendix A). The seven species identified were *N. lotus*, *N. petersiana*, *N. zenkeri*, *N. nouchali* var. *caerulea*, *N. micrantha*, *N. guineensis* and *N. nouchali*.

### 2.3. Genetic Differences and Barcoding Gap Analysis of Candidate DNA Barcoding

Based on the identified species, the genetic divergences of the interspecific and intraspecific variations were calculated for the candidate DNA barcoding using MEGA7. The mean interspecific distances of the four candidate DNA barcodes were 0.159, 0.02, 0.008 and 0.018, respectively. The mean intraspecific distances were 0.004, 0.002, 0.0006 and 0.0015, respectively. The mean interspecific distances of ITS were much greater than the mean interspecific distances of the cpDNA sequences (Table 2).

The ideal DNA barcoding has a significantly smaller intraspecific genetic distances than interspecific distances, with a clear boundary between the two, namely the “barcoding gap” [23]. Although four candidate sequences had overlapping intraspecific and interspecific variations, for individual barcodes, ITS showed the most obvious barcoding gap between intraspecific and interspecific genetic distances, which facilitated the differentiation of *Nymphaea* species. Moreover, the combined sequence (*trnT**-trnF*+*rpl16*) did not show a great advantage over the single sequence (Figure 3). The K2P-based Wilcoxon signed-ranks test clearly reflected divergences between different barcoding markers. The order from large to small was ITS > *trnT**-trnF* > *trnT**-trnF*+*rpl16* > *rpl16* (Table 3).

## 3. Discussion

The study of plant DNA barcoding is moving from the comparison of the performance of different DNA regions to practical applications [17]. In this study, three candidate DNA barcodes were selected for the species identification of 158 *Nymphaea* plants, and 137 populations of seven species were commonly identified by morphological, sequence similarity and phylogenetic analyses.

Since the advent of universal primers for the *trnT-trnF* region, this region has become one of the most widely used regions in plant taxonomy [24]. However, in most cases, only the *trnL* intron and *trnL-trnF* spacer regions have been sequenced, while relatively few analyses have involved the entire *trnT-trnF* region [17,25]. The AT-rich portion of the *trnL* intron (especially the P8 stem–loop region) in *trnT-trnF* provided sufficient information for species identification [26]. Borsch [2] identified 35 *Nymphaea* species based on *trnT-trnF*, with an identification efficiency of 83%. However, the efficiency of identifying *Nymphaea* species using *trnT-trnF* based on the BLAST search or constructing a phylogenetic tree was far less than 83% in this study. In contrast, we found that, in Borsch’s study, most of the samples were one to two per species, and the inadequate sampling resulted in an apparently high resolution. Additionally, the phylogenetic tree based on *trnT-trnF* in this study could only identify *N. petersiana* and *N. nouchali* but could not identify most species of the subg. *Brachyceras*, which was consistent with the results of previous studies [27]. In addition, the identification effect of the *rpl16* and the combined sequence (*trnT-trnF*+*rpl16*) were basically consistent with that of *trnT-trnF* in this study.

Previous studies have found that the evolutionary rate of ITS is three to four times higher than that of the cpDNA regions [18]. Thus, the China Plant BOL Group [28] recommended ITS as a core DNA barcoding for seed plants, and ITS has been widely used as one of the target regions in many phylogenetic and taxonomic studies [29,30,31]. In this study, the ITS sequence length was less than half of the *trnT-trnF*, but the number of variation sites was close to twice that of *trnT-trnF*. Additionally, the variation rate of ITS was as high as 29.73%. Additionally, compared with cpDNA sequences, ITS had the largest intraspecific and interspecific genetic distances, with less overlap between the intraspecific and interspecific variants and an obvious barcoding gap. These characteristics indicated that ITS could be used as a DNA barcode for *Nymphaea*. However, ITS could identify some species of *N. zenkeri*, *N. guineensis*, *N. micrantha* and *N. nouchali* var. *caerulea* by the phylogenetic tree, and a considerable number of species could not be identified, while *trnT-trnF* and *rpl16* could identify some species that could not be identified by ITS. Therefore, it is more appropriate to consider the joint application of ITS as the main DNA barcode of *Nymphaea*, complemented by two chloroplast sequences (*trnT-trnF* and *rpl16*).

In this study, we also tried a new method (ASAP) for species identification, which divides species partitions based on the hierarchical clustering of DNA sequence pairwise genetic distances [22]. However, using ITS and three cpDNA regions can only distinguish well, for the subg. *Nymphaea* was based on the ASAP analysis and cannot accurately identify species. It is possible that the interspecific genetic variants of the subgenera of *Nymphaea* were too low. Therefore, this method is not applicable to the identification of species with small interspecific genetic variants. 

According to the above analysis, ITS, *trnT-trnF*, *rpl16* and *trnT-trnF*+*rpl16* cannot completely distinguish the African *Nymphaea* species. Therefore, there is an urgent need to develop new DNA barcoding or methods to apply to the species identification of African *Nymphaea* species. Hollingsworth et al. [32] proposed that gene capture of nuclear markers and genome skimming will be the focus of future DNA barcoding research. With the rapid development of next-generation sequencing technology and the reduction of sequencing cost, the study of DNA barcoding has moved from DNA fragments to the genomic level. At present, genome skimming has been applied in the DNA barcoding studies of plants such as the genus *Panax* [33], *Taxus* [34] and *Rhododendron* [35], and the discriminatory ability of species has increased significantly. Therefore, continuing to expand the sampling range of *Nymphaea* and applying next-generation sequencing technologies to the identification of *Nymphaea* species are the next research directions.

## 4. Materials and Methods

### 4.1. Sample Collection

In this study, a total of 158 *Nymphaea* populations from seven African countries were sampled in 2018 to 2019 (Figure 4), and each individual was collected at a gap of 5–10 m apart to avoid collecting clonal individuals. The fresh leaves were collected and stored in Ziplock bags containing silica gel at 4 °C until DNA extraction. Since some samples were at a youthful stage or unflowered, it was difficult to identify them from their morphological characters alone. Therefore, we recorded the number for each individual for subsequent identification by molecular data. We identified a total of five species (74 populations) based on morphology, and details of the samples are shown in Appendix A.

### 4.2. DNA Isolation, PCR Amplification and Sequencing

Total genomic DNA was extracted from approximately 30 mg of dried leaves using the MagicMag plant genomic DNA Micro Kit (Sangon Biotech Co., Shanghai, China) according to the manufacturer’s protocol. DNA concentration was checked using the Nanodrop spectrophotometer (Thermo Scientific, Carlsbad, CA, USA), after which it was stored at −20 °C. Then, ITS, *rpl16* and *trnT-trnF* were selected for the DNA barcoding analysis. The *trnT-trnF* region was divided into upstream and downstream segments for amplification. *rps4-5R* and *trnL110R* were used to amplify the *trnT-trnL* spacer [26,36], and primer c and primer f were used to amplify the trnL introns and *trnL-trnF* spacers [24]. The sequences of the four primer pairs are shown in Appendix A. 

PCR amplification was carried out in a 25-μL reaction mixture containing 20 ng of template DNA, 2.5 μL of 10 × Buffer (pH 8.3), 0.5 μL dNTPs, 0.5 μL of each primer, 1 unit of Taq polymerase and DNA-free water. The PCR protocols involved initial denaturation at 94 °C for 5 min, followed by 35 cycles at 94 °C for 40 s, primer-specific annealing temperature (52 °C for ITS4-ITS5 and rpl16 and 56 °C for *rps4-5R*-*trnL110R* and primers c–f) for 1 min and 72 °C for 90 s, a final extension step at 72 °C for 10 min and termination by a final hold at 4 °C. The final PCR products were sequenced using an ABI 3730XL automated sequencer (Applied Biosystems, Foster City, CA, USA).

### 4.3. Sequence Alignment and Data Analysis

The raw sequences were spliced, cut and edited using DNAMAN v.9.0 (Lynnon Corporation, Vaudreuil-Dorion, QC, Canada). In this study, African *Nymphaea* species were identified based on sequence similarity comparison, genetic distance method and phylogenetic analyses. (i) Each edited sequence was submitted to the GenBank database (http://www.ncbi.nlm.nih.gov/genbank, accessed on 23 March 2021). Homology searches were performed using BLAST (http://blast.ncbi.nlm.nih.gov/Blast.cgi, accessed on 23 March 2021) to confirm the identity of the sequences. Due to the few *rpl16* sequences of *Nymphaea* species in the GenBank database, we only performed a homology search of the obtained ITS and *trnT-trnF* sequences with the GenBank database. When the highest similarity between query sequences and matched sequences was over 98%, and ITS and *trnT-trnF* were jointly matched to the same species, the identification of the species was considered successful. (ii) In addition, a total of 51 sequences were downloaded from the GenBank database, including 45 sequences of *Nymphaea* species, 3 sequences of *Euryale ferox* and 3 sequences of *Victoria cruziana* (Appendix A). All sequences obtained were subjected to multiple sequence comparison by the CLUSTALW program in MEGA7 [37] software with visual inspection and manual editing. Species identification was performed using assembled species by automatic partitioning (ASAP) [22], which does not require phylogenetic reconstruction and provides optimal partitioning based on the hierarchical clustering of DNA barcode pairwise genetic distances. The method was applied through an online tool (https://bioinfo.mnhn.fr/abi/public/asap, accessed on 23 March 2021) with default parameters, except for the model set to the Kimura 2 parameter (K2P). (iii) Phylogenetic analyses of the three barcodes were performed using neighbor-joining (NJ) and Bayesian Inference (BI) phylogeny. The NJ tree was constructed using MEGA7 software based on the K2P model with the bootstrap set to 1000. The best-fit model of the nucleotide substitution was determined by ModelFinder [38] with Akaike Information Criterion (AIC). BI phylogeny was inferred using MrBayes v.3.2 [39], running for 10 million generations and an initial 25% of sampled data as the burn-in.

### 4.4. Genetic Divergence Analysis

The intraspecific and interspecific genetic distances of the species were calculated based on the K2P model using MEGA7. To estimate the barcoding gap, the minimum interspecific and maximum intraspecific distances were compared for each barcode [40]. Then, we used the Wilcoxon signed-rank test based on the K2P model to assess the significance of intraspecific distances against interspecific divergences for each pair of barcodes in SPSS V21.0.

## Figures and Tables

**Figure 1 plants-11-02431-f001:**
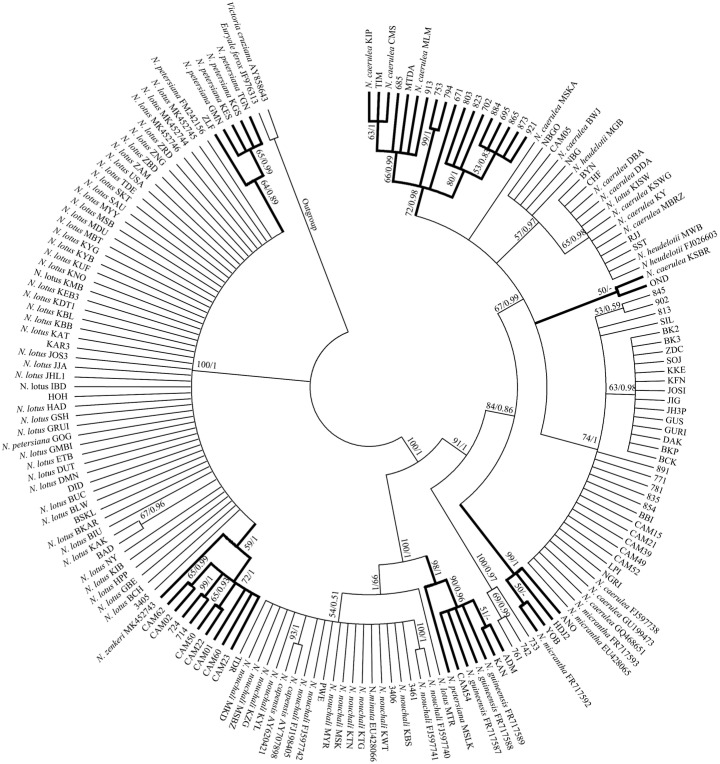
Phylogenetic tree constructed based on 179 ITS sequences of *Nymphaea*. Bootstrap values (>50) and posterior probabilities (>0.5) based on the NJ and BI analyses are shown above the branches. The rough branches mean identified species by ITS tree.

**Figure 2 plants-11-02431-f002:**
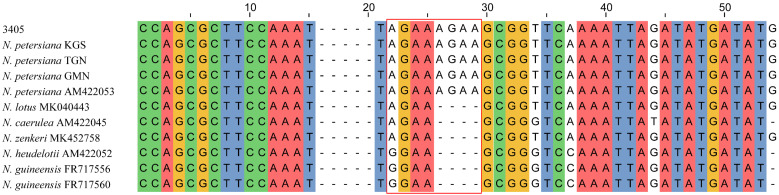
The partial sites in the *trnL-trnF* spacer region of some sequences. The red box means “AGAA”-SSR in *N. petersiana*.

**Figure 3 plants-11-02431-f003:**
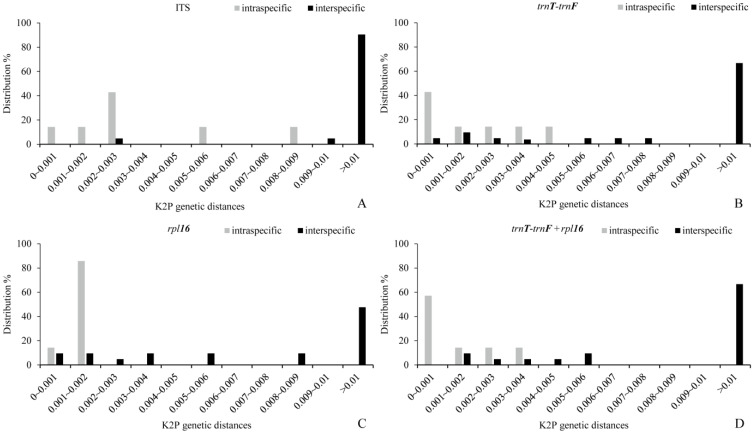
Relative distribution of inter- and intraspecific distances of each sequence of *Nymphaea*. (**A**) ITS, (**B**) *trnT-trnF*, (**C**) *rpl16* and (**D**) *trnT-trnF+rpl16*.

**Figure 4 plants-11-02431-f004:**
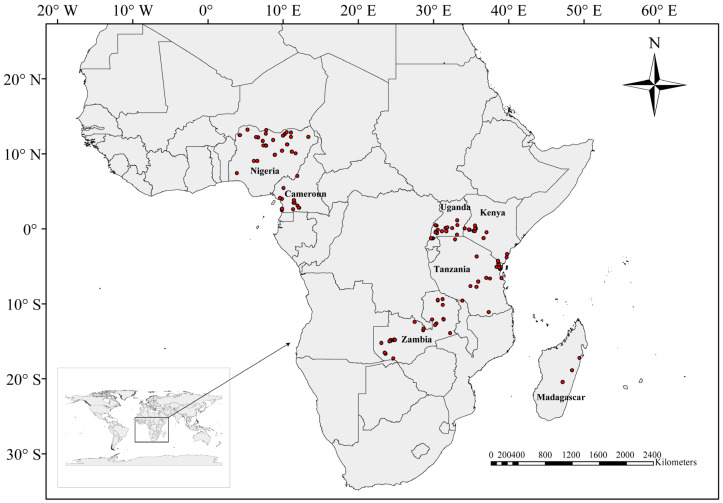
Sampling sites in seven African countries.

**Table 1 plants-11-02431-t001:** Characteristics of the DNA barcode sequence and success rate of PCR amplification and sequencing.

Sequences	Percentage PCR Success (%)	Percentage Sequencing Success (%)	Aligned Sequence Length (bp)	GC Content (%)	Variable Sites	Percentage Variable (%)	Parsimony Informative Sites
ITS	100	99.37	731	51.8	212	29.73	196
*trnT-trnF*	98.1	96.2	1507	36.61	126	8.36	70
*rpl16*	100	100	766	36.1	34	4.44	15

**Table 2 plants-11-02431-t002:** Parameters of the interspecific and intraspecific variations of each DNA barcoding.

Sequences	No. of Samples	The Range of Interspecific Distance	The Range of Intraspecific Distance	Average Interspecific Distance	Average Intraspecific Distance
ITS	144	0.002–0.261	0–0.009	0.159 ± 0.016	0.004 ± 0.001
*trnT-trnF*	141	0.002–0.033	0.001–0.004	0.02 ± 0.003	0.002 ± 0.0006
*rpl16*	144	0–0.014	0–0.001	0.008 ± 0.003	0.0006 ± 0.0003
*trnT-trnF+* *rpl16*	141	0.001–0.027	0.001–0.003	0.018 ± 0.003	0.0015 ± 0.0004

**Table 3 plants-11-02431-t003:** Wilcoxon signed-rank tests ^1^ among the interspecific divergence of each DNA barcode.

W+	W−	RELATIVE Ranks		N	*p*-Value≤	Result
		W+	W−			
ITS	*trnT-trnF*	206	5	21	0.001	ITS > *trnT-trnF*
ITS	*rpl16*	206	4	21	0.001	ITS > *rpl16*
ITS	*trnT-trnF+rpl16*	226	5	21	0.001	ITS > *trnT-trnF+rpl16*
*trnT-trnF*	*rpl16*	203.5	27.5	21	0.003	*trnT-trnF* > *rpl16*
*trnT-trnF*	*trnT-trnF+rpl16*	187	3	21	0.001	*trnT-trnF* > *trnT-trnF+rpl16*
*rpl16*	*trnT-trnF+rpl16*	28.5	202.5	21	0.003	*rpl16* < *trnT-trnF+rpl16*
ITS	*trnT-trnF*	206	5	21	0.001	ITS > *trnT-trnF*
ITS	*rpl16*	206	4	21	0.001	ITS > *rpl16*
ITS	*trnT-trnF+rpl16*	226	5	21	0.001	ITS > *trnT-trnF+rpl16*

^1^ The symbols “W+” and “W−” represent the sums of the positive and negative values in the signed-rank column, respectively. Symbol “>” is used in the interspecific divergence for one barcoding marker significantly exceeding that of another barcoding marker.

## Data Availability

The DNA sequences are available in the NCBI GenBank (accession numbers ITS: MW798802–MW798955, *trnT-trnF*: MW809732–MW809883 and *rpl16*: MW809884–MW810041).

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
