# Peer review of "Molecular Identification of African Nymphaea Species (Water Lily) Based on ITS, trnT-trnF and rpl16"

_plants, 2022, doi:10.3390/plants11182431_

Round 1

Reviewer 1 Report

The authors apply the combination of ITS and trnT-trnF (or rpl16) as the DNA barcoding of Nymphaea, which will enrich the DNA barcoding database of Nymphaea and provide a reference basis for studying taxonomy, phylogenetics and evolutionary origin of Nymphaea. But the English writing should be improved. Additionally, I have only few minor comments:

Line 38  change 'ability to purify water bodies' to 'water purification capacity'

Line 39 change taxa to lineage

Line 45 change 'It is a pity because' to 'Though'

Author Response

The authors apply the combination of ITS and trnT-trnF (or rpl16) as the DNA barcoding of Nymphaea, which will enrich the DNA barcoding database of Nymphaea and provide a reference basis for studying taxonomy, phylogenetics and evolutionary origin of Nymphaea. But the English writing should be improved. Additionally, I have only few minor comments.

Response:

Thanks for your comments. We have corrected our MS according to your suggestions. Please see below.

1. Line 38  change 'ability to purify water bodies' to 'water purification capacity'

Response:

We agree with your suggestion. We have modified “ability to purify water bodies” to “water purification capacity”.

Line 39 change taxa to lineage

Response:

We agree with your suggestion. We have modified “taxa” to “lineage”.

Line 45 change 'It is a pity because' to 'Though'

Response:

We agree with your suggestion. We have modified “It is a pity because” to “Though”.

Reviewer 2 Report

The work is of a high standard and of great interest. The research aims to identify suitable tools to clarify the taxonomy among the species belonging to the genus Nynphaea. The results obtained lay the groundwork for greater clarity in the classification and history of evolution of these species.

The methodology used is clear.

The discussion is well structured and the comparison with previous works clarifies the results obtained

Author Response

The work is of a high standard and of great interest. The research aims to identify suitable tools to clarify the taxonomy among the species belonging to the genus Nynphaea. The results obtained lay the groundwork for greater clarity in the classification and history of evolution of these species.

The methodology used is clear.

The discussion is well structured and the comparison with previous works clarifies the results obtained.

Response:

Thanks for your comments. 
